# A Study on Behavioral Intentions in the Field of Eco-Friendly Drone Food Delivery Services: Focusing on Demographic Characteristics and Past Experiences

Jinsoo Hwang [1], Kyuhyeon Joo [1] and Joonho Moon [2,*]

1   The College of Hospitality and Tourism Management, Sejong University, Seoul 143747, Republic of Korea
2   Department of Tourism Administration, Kangwon National University,
    Chooncheon 24341, Republic of Korea
*   Correspondence: joonhomoon0412@gmail.com

**Abstract:** Drones operate on electric batteries and not on gasoline, so the eco-friendly role of drones has recently attracted a lot of attention. Thus, this study was designed in order to investigate differences in behavioral intentions, such as intention to use, word-of-mouth, and willingness to pay more, according to demographic characteristics and past experiences in the field of eco-friendly drone food delivery services. Data were collected from 422 potential consumers of eco-friendly drone food delivery services in South Korea. The data analysis results indicated that females are more willing to pay extra than males are, respondents who were in their 50s had higher word-of-mouth intention than other generations, marital status showed significant differences in willingness to pay more and intentions to use, and there was a difference in willingness to pay more and word-of-mouth with regards to monthly income. In addition, respondents who had previously heard of drone food delivery services had higher averages with willingness to pay more and intentions to use as opposed to respondents who had not heard of them, and respondents who had experience controlling drones were willing to pay additional fees when they used eco-friendly drone food delivery services. The results of this study would be a great assistance for executives who will operate eco-friendly drone food delivery services.

**Keywords:** eco-friendly; behavioral intentions; drones; demographic characteristics; food delivery

## 1. Introduction

The food service industry has experienced significant changes in recent times that are due to advancements in food technology, which has had a profound effect on consumers [1]. Drone-based food delivery services have emerged as one of the most significant topics of discussion in the fourth industrial revolution among various new food technologies [2], because drones are regarded as becoming central to the functions of diverse business operations, such as disaster management, safety inspections, transport services, law enforcement, and control surveillance [3].

The important role of drones in the restaurant industry is no exception. Drone-based services are expected to play a significant role in the food service industry by providing several important benefits. The most significant advantage of drone-based food delivery services is the ability to save time by avoiding traffic congestion and delivering food to any location without restrictions [4]. This service also helps companies save on delivery worker wages [5]. In addition, the most important advantage of drones is their eco-friendly role with regards to global warming, which is caused by the use of fossil fuels and has been identified as a primary factor that contributes to droughts, flooding, heat waves, and desertification in numerous regions across the globe, and therefore, many countries have been trying to focus on environmental protection in recent years [6,7]. According to Environmental Technology [8], the use of drones for food deliveries is beneficial for the environ-

ment, because they run on electricity, which contributes to environmental protection. This means that drone-based food delivery services are a critical way to protect the environment.

This study focused on consumer behavioral intentions in the field of eco-friendly drone food delivery services. It is widely known that behavioral intentions are a key construct in the field of business and management [9,10], and can be defined as an individual's subjective likelihood or willingness to engage in a particular behavior [11]. The most important reason above all to study behavioral intentions is that behavioral intentions and actual behavior are very closely related [12], which means that if consumers have a high level of behavioral intention, they are more likely to have actual purchase intentions. For this reason, many previous studies consistently examined the predictors of behavioral intentions [13–15].

Previous research shows that demographic characteristics and past experiences play a significant role in consumer behavior [16,17]. According to Kral, Janoskova, Lazaroiu, and Suler [18], demographic characteristics such as gender, age, education, and income have been identified as important factors with regards to understanding consumer behavior. These demographic variables are especially crucial in the context of green industries [19,20], which suggests that research on environmental issues must consider demographic characteristics in order to accurately assess pro-environmental behavior among consumers. In addition, past experiences are also a key factor with regards to understanding consumer behavior in the green industry [21,22].

In addition, some studies have been conducted in the field of drone food delivery services. For example, Hwang, Kim, and Kim [23] applied the concept of motivated consumer innovativeness in drone food delivery services. Hwang, Kim, and Gulzar [24] examined how to form behavior intentions based on the theory of planned behavior in the context of drone food delivery services. More recently, Hwang, Kim, and Lee [25] investigated consumer innovativeness in the context of drone food delivery services. Unlike previous studies, this study explored differences in behavioral intentions, such as intention to use, word-of-mouth, and willingness to pay more, according to demographic characteristics and past experiences in the field of eco-friendly drone food delivery services for the first time. This is a significant difference from previous studies and can be said to be an important theoretical implication of this study.

In summary, this study was designed in order to examine differences in behavioral intentions, which include intention to use, word-of-mouth, and willingness to pay more, according to demographic characteristics and past experiences in the context of eco-friendly drone food delivery services. The results of this study would be a great assistance to executives who will operate eco-friendly drone food delivery services in the future.

## 2. Literature Review

### 2.1. Drones

Drones are unmanned aerial vehicles that originated from military applications during World War I [26], and the utilization of drones has been greatly expanded across many different sectors [27,28]. The innovativeness of drones was recognized in delivery services when drone food delivery services became a reality in 2016, when Domino's Pizza delivered pizzas to a customer's door in New Zealand [5]. Drone food delivery services are not completely commercialized in business practices today, but the potential usage of drones in food delivery services is steadily gaining momentum. Drones above all are free from traffic congestion, which enables food to be delivered a lot faster [27,29].

Drones are also, more importantly, recognized as a pro-environmental delivery mode, because they reduce carbon dioxide ($CO_2$) and greenhouse gas emissions [30]. Furthermore, Doole et al. [31], via a cost-analysis study, demonstrated that the cost range of drone-based food delivery services is only half of the cost of using electric bike-based delivery methods. Furthermore, Jaramillo et al. [32], via the application of the analytical hierarchy process (AHP), determined that reducing time and energy are key success factors of

drone food delivery services. Drone-based delivery services have more recently been addressed as an effective solution to the lockdowns which were caused by the coronavirus pandemic [33]. This means that autonomous drones can be used in order to cope with COVID-19 by carrying food to self-isolated individuals. Drones are more popular now than ever as a means of offering additional value in the food delivery industry.

### 2.2. Behavioral Intentions towards Drone Food Delivery Services

The concept of behavioral intentions refers to "a stated likelihood to engage in a behavior" [11] (p. 28). Scholars generally agree that there are three main components of behavioral intentions, which include intentions to use, word-of-mouth intentions, and willingness to pay more [34,35].

First, intentions to use can be defined as "the degree that a person has formulated conscious plans to perform or not perform some specified future behavior" [36] (p. 214). Intentions to use are established based on a customer's positive assessment of a product or service [37,38]. Intentions to use additionally have a direct impact on actual consumption, so previous studies endeavored to determine the factors that influence intentions to use [39–41].

Second, according to Westbrook [42] (p. 261), word-of-mouth intentions are defined as "informal communication directly at other consumers about the ownership, usage, or characteristics of particular goods and services and/or their sellers". Harrison-Walker [43] (p. 63) similarly defined word-of-mouth intentions as "informal person to person communication between a perceived noncommercial communicator and a receiver regarding a brand, a product, an organization, or a service". Word-of-mouth is regarded as rather fair and unbiased, because it is based on interactions among consumers themselves. It creates more powerful marketing effects, according to Mourali, Laroche, and Pons [44].

Third, willingness to pay more refers to the probability of consumers' own volition to spend more for a product or a service [44]. It has also been often examined as an indicator of behavioral intentions, because consumers have a tendency to pay more for a better product quality, which increases consumption value [45]. In addition, previous research revealed that perceived quality helps to enhance willingness to pay more for a product or service [46,47]. This implies that customers are more likely to spend additional money when they perceive high levels of quality. Willingness to pay more is crucial for a company with regards to increasing sales and revenue [48,49].

### 3. Methodology

### 3.1. Measurement

The constructs in the research model were measured using scales that included multiple items. The items were borrowed from previous research and were modified for this study. Behavioral intentions included three sub-dimensions, namely, intentions to use, word-of-mouth, and willingness to pay more. Three items were borrowed from Zeithaml, Berry, and Parasuraman [50] in order to assess intention to use. Word-of-mouth intentions were measured using three items that were obtained from Hennig-Thurau, Gwinner, and Gremler [51], and willingness to pay more was measured using three items from Han et al. [34]. All the items were evaluated using a Likert scale which ranged from strongly disagree to strongly agree. In addition, we made some modifications to the original items in order to better fit the context of eco-friendly drone food delivery services. The initial questionnaire was created based on the measurement items, and two expert groups, which included faculty members and drone experts, then carefully reviewed the questionnaire.

We conducted a pilot test using 50 actual restaurant patrons in order to check the reliability of all the measurement items. Eco-friendly drone food delivery services are being conducted on a trial basis in South Korea, but they have not been commercialized enough in order to be widely used by ordinary people. The respondents may have a poor understanding of the services, so roughly about two minutes' worth of newspaper articles that clearly describe the eco-friendly drone food delivery services were provided to the respon-

dents prior to the start of the survey. As a result, the Cronbach's alpha values for the nine constructs were higher than 0.7, which supported the fact that all the constructs have high levels of reliability [52].

### 3.2. Hypothesis Development

Demographic characteristics have long been used in order to better understand consumer behavior in the field of social sciences [53–55]. Demographic characteristics in particular have an important meaning with regards to eco-friendly research. According to Han, Hsu, Lee, and Sheu [56], eco-friendly behavioral intentions are very different depending on the demographic characteristics of consumers, so eco-friendly research must consider the demographic characteristics of consumers. Existing empirical studies also show the importance of demographic characteristics in eco-friendly research. For example, Kwon and Ahn [19] investigated the influence of respondents' demographic characteristics on behavioral intention toward green hotels. They discovered that age and education level are important factors that affect behavioral intention. Wang, Wong, and Narayanan [20] additionally explored the effect of demographic characteristics on consumers' green purchase attitudes and behavioral intention in the green hotel industry. They suggested that there were significant differences in green purchase attitudes with regards to age and income, whereas there was a significant difference in green behavioral intention with regards to education. Kim, Joo, and Hwang [57] more recently discovered that there were partial statistical differences in four dimensions of the internal environmental locus of control, namely, green consumers, environmental activists, environmental advocates, and recyclers, with regards to gender, age, education, marital status, and monthly income.

Furthermore, past behavior shows consumers' habits and preferences, so identifying past behavior is very important with regards to understanding their behavior intentions [56]. It is particularly important to understand past experiences in green research, because green behavior can be predicted from past experiences [58]. Previous research also supports the significance of behavioral intentions. For example, Han and Hwang [59] developed a research model in order to find the role of past experiences in the formation of behavioral intentions in an environmentally responsible cruise context. They revealed that past experiences affect behavioral intentions. Han, Hwang, Kim, and Jung [60] examined the effect of past behavior on pro-environmental intention to revisit in a lodging context. They showed that past behavior is a key predictor of pro-environmental intention to revisit. Han, Kim, and Lee [61] investigated how past experiences affect museum visitors' pro-environmental behavior. They showed that past experiences assist in forming museum visitors' pro-environmental behavior. The following hypotheses are postulated as a result of the above arguments.

**Hypothesis 1 (H1).** There are differences in behavioral intentions based on demographic characteristics.

**Hypothesis 1a (H1a).** There are differences in behavioral intentions based on gender.

**Hypothesis 1b (H1b).** There are differences in behavioral intentions based on age.

**Hypothesis 1c (H1c).** There are differences in behavioral intentions based on level of education.

**Hypothesis 1d (H1d).** There are differences in behavioral intentions based on marital status.

**Hypothesis 1e (H1e).** There are differences in behavioral intentions based on the level of monthly income.

**Hypothesis 2 (H2).** There are differences in behavioral intentions based on past experiences.

**Hypothesis 2a (H2a).** There are differences in behavioral intentions based on whether respondents have heard of drone delivery services.

**Hypothesis 2b (H2b).** There are differences in behavioral intentions based on whether respondents have controlled a drone.

**Hypothesis 2c (H2c).** There are differences in behavioral intentions based on whether respondents currently own a drone.

*3.3. Data Collection*

This study employed an online panel survey, which was administered by a market research firm in South Korea for the main data collection. The survey was distributed to individuals who had used food delivery services within the past six months. The participants were presented with a newspaper article, which was similar to the pretest, that emphasized the role of drone food delivery services in environmental protection, before responding to the questionnaire. A total of 4525 panelists were invited via email to participate in the survey, and 442 panelists responded to it. A sample of 436 responses was used for statistical analysis after inappropriate responses were eliminated, which included responses with multicollinearity problems and visual inspections.

**4. Data Analysis**

*4.1. Respondent Profile*

Table 1 displays the demographic characteristics of the survey participants. Overall, 209 of the respondents were male (47.9%), and 227 of the respondents were female (52.1%). In addition, most of the respondents were in their 30s (33.3%), which was followed by respondents in their 40s (26.1%). The majority of respondents reported a high level of education, which included 63.8% ($n = 278$) who indicated that they held a bachelor's degree. Overall, 51.4% of participants ($n = 224$) were single. Lastly, 28.0% of participants ($n = 122$) reported a monthly household income between USD 1001 and USD 2000. The mean age of respondents was 37.81 year, and approximately 51.4% of them were single. According to the Korean Statistical Information Service, there are more women than men, and the highest proportion of adults with a bachelor's degree in education [62], which is similar to the respondents' properties.

**Table 1.** The respondents' profile ($n = 436$).

| Variable | $n$ | Percentage |
| --- | --- | --- |
| **Gender** | | |
| Male | 209 | 47.9 |
| Female | 227 | 52.1 |
| **Age** | | |
| 20s | 109 | 25.0 |
| 30s | 145 | 33.3 |
| 40s | 114 | 26.1 |
| 50s | 68 | 15.6 |

**Table 1.** *Cont.*

| Variable | *n* | Percentage |
|---|---|---|
| **Education Level** | | |
| High school diploma | 50 | 11.5 |
| Associate's degree | 69 | 15.8 |
| Bachelor's degree | 278 | 63.8 |
| Graduate degree | 39 | 8.9 |
| **Marital Status** | | |
| Single | 224 | 51.4 |
| Married | 208 | 47.7 |
| Others (divorced and widow/widower) | 4 | 0.9 |
| **Monthly income** | | |
| Less than USD 1000 | 64 | 14.7 |
| USD 1001~USD 2000 | 122 | 28.0 |
| USD 2001~USD 3000 | 119 | 27.3 |
| USD 3001~USD 4000 | 55 | 12.6 |
| USD 4000~USD 5000 | 37 | 8.5 |
| More than USD 5001 | 39 | 8.9 |

*4.2. Respondents' Information Related to Drones*

Table 2 shows respondents' information with regards to drones. First, 237 respondents (54.4%) indicated that they had heard of drone delivery services. Second, 117 respondents (26.8%) revealed that they have personal experience controlling drones. Third, 79 respondents (18.1%) revealed that they currently own a drone or have previously owned one.

**Table 2.** Respondents' past experiences with drones (*n* = 436).

| Variable | *n* | Percentage |
|---|---|---|
| **Have you ever heard of drone delivery services?** | | |
| Yes | 237 | 54.4 |
| No | 199 | 45.6 |
| **Have you ever controlled a drone yourself?** | | |
| Yes | 117 | 26.8 |
| No | 319 | 73.2 |
| **Do you currently own a drone or have you ever owned one before?** | | |
| Yes | 79 | 18.1 |
| No | 357 | 81.9 |

*4.3. Results of Principal Component Analysis*

Table 3 shows the measurement scales and the results of the principal component analysis (The original measurement scales in the Korean language were presented in Appendix A). Principal component analysis was conducted in order to evaluate the underlying dimensions of behavioral intentions, which resulted in a unidimensional model with three factors that included willingness to pay more, word-of-mouth, and intentions to use. Each factor had an eigenvalue greater than 1.0, and the validity of the factor model was confirmed by having a Kaiser–Meyer–Olkin (KMO) value of 0.880 and a statistically significant Bartlett's test of sphericity of $p < 0.001$. The model explained 93.459% of the variance, which included factor loadings above 0.849 as well as Cronbach's alpha values above 0.7 for each construct.

**Table 3.** The results of principal component analysis for behavioral intentions.

| Variables (Mean and Standard Deviation) | Factor Loading | Eigenvalue | Explained Variance | Cronbach's α |
|---|---|---|---|---|
| **Willingness to pay more (3.51 and 1.40)** | | 2.917 | 32.406 | 0.973 |
| I am likely to spend extra in order to use drone food delivery services. | 0.929 | | | |
| It is acceptable to pay more for drone food delivery services. | 0.928 | | | |
| I am likely to pay more for drone food delivery services. | 0.918 | | | |
| **Word-of-mouth (4.61 and 1.33)** | | 2.778 | 30.866 | 0.953 |
| I am likely to recommend drone food delivery services to others. | 0.878 | | | |
| I am likely to encourage others to use drone food delivery services. | 0.872 | | | |
| I am likely to say positive things about drone food delivery services to others. | 0.858 | | | |
| **Intentions to use (4.42 and 1.35)** | | 2.717 | 30.187 | 0.968 |
| I am likely to use drone food delivery services when ordering food. | 0.871 | | | |
| I will use drone food delivery services when ordering food. | 0.862 | | | |
| I am willing to use drone food delivery services when ordering food. | 0.849 | | | |

Notes: KMO measure of sampling adequacy = 0.880, Bartlett's test of sphericity ($p < 0.001$), and the total explained variance = 93.459%.

*4.4. Results of t-tests and one-way ANOVA for Demographic Characteristics*

*t*-tests and one-way ANOVA were conducted in order to examine whether there were differences in behavioral intentions based on demographic characteristics, and are shown in Table 4. The results of the *t*-tests revealed a significant difference in willingness to pay more with regards to gender. Females are more willing to pay extra than the males are. The one-way ANOVA results indicated significant differences in word-of-mouth with regards to age. It was discovered that respondents who were in their 50s had higher word-of-mouth intention than other generations. Furthermore, marital status showed significant differences with regards to willingness to pay more and intentions to use. Lastly, there was a difference in willingness to pay more and word-of-mouth with regards to monthly income. The averages of willingness to pay more and word-of-mouth were higher in the group with relatively high monthly income, namely, USD 4000~USD 5000 and more than USD 5001.

**Table 4.** The results of *t*-tests and one-way ANOVA for demographic characteristics.

| Gender | Male | | | Female | *t*-Value | *p*-Value |
|---|---|---|---|---|---|---|
| Willingness to pay more | 3.37 | | | 3.64 | 2.027 | 0.043 ** |
| Word-of-mouth | 4.50 | | | 4.70 | 1.529 | 0.127 |
| Intentions to use | 4.41 | | | 4.42 | 0.110 | 0.913 |
| **Age** | 20s | 30s | 40s | 50s | *F*-value | *p*-value |
| Willingness to pay more | 3.46 | 3.42 | 3.55 | 3.70 | 0.686 | 0.561 |
| Word-of-mouth | 4.74 | 4.40 | 4.52 | 4.99 | 3.677 | 0.012 ** |
| Intentions to use | 4.40 | 4.34 | 4.32 | 4.75 | 1.679 | 0.171 |
| **Education** | High school diploma | Associate's degree | Bachelor's degree | Graduate degree | *F*-value | *p*-value |
| Willingness to pay more | 3.42 | 3.38 | 3.54 | 3.61 | 0.395 | 0.757 |
| Word-of-mouth | 4.48 | 4.49 | 4.66 | 4.60 | 0.440 | 0.724 |
| Intentions to use | 4.52 | 4.28 | 4.43 | 4.38 | 0.342 | 0.795 |

**Table 4.** *Cont.*

| Marital status | Single | Married | Others | *F*-value | *p*-value |
|---|---|---|---|---|---|
| Willingness to pay more | 3.41 | 3.59 | 4.75 | 2.409 | 0.091 * |
| Word-of-mouth | 4.58 | 4.62 | 5.75 | 1.535 | 0.217 |
| Intentions to use | 4.42 | 4.37 | 6.25 | 3.819 | 0.023 ** |

| Monthly income (Unit: USD) | Less than USD 1000 | USD 1001~USD 2000 | USD 2001~USD 3000 | USD 3001~USD 4000 | USD 4000~USD 5000 | More than USD 5001 | *F*-value | *p*-value |
|---|---|---|---|---|---|---|---|---|
| Willingness to pay more | 3.32 | 3.45 | 3.41 | 3.31 | 4.03 | 4.07 | 2.947 | 0.013 ** |
| Word-of-mouth | 4.68 | 4.61 | 4.50 | 4.30 | 4.71 | 5.11 | 1.978 | 0.081 * |
| Intentions to use | 4.34 | 4.37 | 4.35 | 4.27 | 4.57 | 4.95 | 1.599 | 0.159 |

Notes: * $p < 0.1$ and ** $p < 0.05$.

### 4.5. The Results of the t-tests for Respondents' Past Experiences with Drones

*t*-tests were also performed in order to explore whether there were differences in behavioral intentions based on respondents' information related to drones, and are shown in Table 5. The data analysis showed that respondents who had previously heard of drone food delivery services had higher averages of willingness to pay more and intentions to use, as opposed to respondents who had not heard of them. In addition, respondents who had experience controlling drones were willing to pay additional fees when they used eco-friendly drone food delivery services.

**Table 5.** The results of *t*-tests for respondents' past experiences with drones.

| Have you ever heard of drone delivery services? | Yes | No | *t*-value | *p*-value |
|---|---|---|---|---|
| Willingness to pay more | 3.68 | 3.30 | 2.863 | 0.004 ** |
| Word-of-mouth | 4.67 | 4.53 | 1.090 | 0.276 |
| Intentions to use | 4.53 | 4.28 | 1.926 | 0.055 * |
| **Have you ever controlled a drone yourself?** | **Yes** | **No** | ***t*-value** | ***p*-value** |
| Willingness to pay more | 3.80 | 3.40 | 2.677 | 0.008 ** |
| Word-of-mouth | 4.62 | 4.56 | 0.437 | 0.662 |
| Intentions to use | 4.52 | 4.38 | 0.969 | 0.333 |
| **Do you currently own a drone or have you owned one before?** | **Yes** | **No** | ***t*-value** | ***p*-value** |
| Willingness to pay more | 3.66 | 3.48 | 1.066 | 0.287 |
| Word-of-mouth | 4.42 | 4.65 | 1.360 | 0.175 |
| Intentions to use | 4.53 | 4.39 | 0.837 | 0.403 |

Notes: * $p < 0.1$ and ** $p < 0.05$.

## 5. Discussion and Implications

### 5.1. Theoretical Implications

First of all, the present paper is the first investigation of differences in behavioral intentions based on demographic factors and past experience. Although extant studies examined consumer behavior in the context of drone food delivery services (e.g., [4,13,29,39,41]), they overlooked differences in behavioral intentions according to demographic characteristics, and there was also no study focusing on past experience. Thus, this study focused on demographic factors and past experience. The study developed the questionnaire by reviewing it with faculty members and drone experts, and successfully identified statistical differences in behavioral intentions based on demographic factors and past experiences.

More specifically, there were significant differences in willingness to pay more according to gender. This is in line with the study on sex differences in social behavior by Eagly [63]. Females are more aware of environmental issues, and they tend to behave in a more environmentally friendly way, so they are more willing to pay extra for products and services compared to males. Second, there were significant differences in word-of-mouth according to age. It was found that respondents in their 50s had higher word-of-mouth intentions than other generations. Furthermore, the previous literature also in-

dicates that older consumers have more ecological concerns, and they tend to advocate pro-environmental behavior more than the younger generations [64,65]. Third, marital status showed significant differences in willingness to pay more and intentions to use. It was discovered that respondents in the other group, which included divorced and widows/widowers, had higher values than single and married people. However, the results should be interpreted with caution, because there were only four respondents in this group. Fourth, there was a significant difference in willingness to pay more and word-of-mouth with regards to monthly income. The averages of willingness to pay more and word-of-mouth were higher in the group with a relatively high monthly income. This is also in line with the study by Gatersleben et al. [66]. They discovered that consumers with high income levels are more concerned about environmental problems, and they tend to behave pro-environmentally. These discussions present theoretical contributions which imply that the effects of demographic characteristics on behavioral intentions in the drone food delivery service context are associated with pro-environmental behavior.

In addition, the study also successfully identified differences in behavioral intentions based on respondents' past experiences with drones. The results indicated that respondents who had previously heard of drone food delivery services had a higher willingness to pay extra and intentions to use than individuals who had not heard of drone food delivery services. It was also revealed that respondents who had experience controlling drones were willing to pay additional fees when using eco-friendly drone food delivery services. These findings are in line with previous studies which indicated that past behavior is a key predictor of pro-environmental behavior [50,52–55]. This study consequently presents empirical evidence of the effect of past experiences on behavioral intentions in the context of drone food delivery services for the first time.

However, there are no statistical differences in behavioral intentions according to education level and whether consumers have owned a drone or not. Past studies stated that information and education about environmental protection can foster pro-environmental behavior [66]. As the issue of environmental pollution has become widely disseminated online, there might therefore not be differences in behavioral intentions according to education levels. In addition, consumers can directly/indirectly experience drones through YouTube videos, VR (virtual reality), and trial booths in the exhibition, even if they do not own a drone. Thus, there might be no difference in behavioral intentions based on whether or not they own a drone, unlike whether or not they had heard of or controlled a drone. These discussions imply that consumer behavior can change with technology development for experience and information.

*5.2. Practical Implications*

The present study revealed that people who had previously controlled drones were more willing to pay extra toward eco-friendly drone food delivery services than people who had not controlled a drone. This also indicated that people who had heard of drone food delivery services were more intent to use the services than people who had not heard of them, so this study suggests an experience marketing strategy. For instance, marketers at a food delivery service company can plan a green campaign that can promote the company's green initiative by utilizing drones instead of motorbike delivery services which cause environmental pollution. The study also suggests an experiential event where consumers acquire specific green goods, such as eco-tumblers and eco-bags, by controlling drones. This type of campaign would have a crucial role with regards to forming memorable experiences controlling drones and with the promotion of services. Companies can also conduct target marketing that is based on the demographic factors that are identified in this study. Many people in their 50s and people who are older enjoy controlling drones as a hobby or leisure activity [67]. This study also revealed that people in their 50s had higher word-of-mouth intentions than other generations. It can be interpreted that marketers should target consumers who are in their 50s when they plan a green campaign, which is suggested above, in order to maximize the promotional effect of the campaign.

Females were also more willing to pay extra toward eco-friendly drone food delivery services than males were, so companies should consider targeting females as a key demographic variable when they plan marketing strategies for drone food delivery services. For instance, managers can consider prioritizing coffee shops or salad cafés, which have a higher portion of female customers than male customers, when they select brands in order to introduce drone food delivery services. The brands should be prioritized as high-end brands as opposed to low-cost brands, because the study revealed that the averages of willingness to pay more and word-of-mouth were higher in respondents that had a relatively high monthly income. Furthermore, an event can be conducted that considers that females were more willing to pay extra toward eco-friendly drone food delivery services than males. A green event with brand collaborating can be planned that involves a condition where, if customers pay an additional fee of approximately USD 2, the brand stores provide food in eco-friendly multi-use containers. Product placement (PPL), which is an indirect advertisement strategy in Korean media, can also be planned that shows a heroine engaging in the event in a drama that has a high proportion of female viewers. This type of advertisement would play a crucial role with regards to publicizing eco-friendly drone food delivery services to female consumers.

## 6. Conclusions

The present study was designed in order to investigate differences in behavioral intentions that included intention to use, word-of-mouth, and willingness to pay more which were based on demographic characteristics and past experiences in the field of eco-friendly drone food delivery services. Data were collected from 422 potential consumers of eco-friendly drone food delivery services in Korea. The results indicated that (1) there were significant differences in willingness to pay more which were based on gender, marital status, and monthly income, whether respondents had heard of drone delivery services or not, as well as whether respondents had ever controlled a drone themselves or not. (2) There were significant differences in word-of-mouth based on age and monthly income, and (3) there were significant differences in intentions to use based on marital status and whether respondents had ever heard of drone delivery services or not. These findings present theoretical contributions, because the effects of demographic characteristics on behavioral intentions in the drone food delivery service context are associated with pro-environmental behavior. It also presents empirical evidence of the effect of past experiences on behavioral intentions in the context of drone food delivery services for the first time. Lastly, the study also presents practical suggestions, such as experience marketing and PPL advertisements.

Nonetheless, the limitations of this study should not be overlooked. First, the generalizability of the findings is limited, because the present study collected data only from Korean respondents. In addition, the median age in South Korea is about 44 years old [62], so the results may not be representative of the country studied. It is necessary to design a sample that reflects the demographic population of the country studied. Furthermore, respondents did not represent actual users of drone food delivery services, because they have not yet been fully commercialized in Korea. Future studies should consider a data-collection method which can improve generalizability, such as cross-cultural studies and research on actual users. In addition, this study only focused on difference analysis as a one-way method. AnswerTree algorithms can provide in-depth results by splitting nodes. It is also possible to investigate the factors predicting behavioral intentions and perform regression analysis. For instance, the risks that consumers are concerned about with services weakened their behavioral intentions [13,29]. On the other hand, individuals' innovativeness strengthens their behavioral intentions [15]. The present study might have overlooked the effects of these types of variables, so this suggests further research design of a framework that encompasses both negative and positive factors with demographic factors. Lastly, the study only focused on consumers' personal conditions, which included demographic factors and past experiences. Behavioral intentions can be fostered by social

factors, such as norms and social influence [4,39]. The study suggests that future research consider differences in social motives in the context of drone food delivery services.

**Author Contributions:** Conceptualization, J.H. and J.M.; methodology, J.H. and J.M.; software, J.H.; validation, J.H.; formal analysis, K.J.; investigation, J.M., K.J. and J.M.; resources, J.M.; data curation, K.J. and J.M.; writing—original draft preparation, J.H. and J.M.; writing—review and editing, J.H. and K.J. All authors have read and agreed to the published version of the manuscript.

**Funding:** This research received no external funding.

**Institutional Review Board Statement:** Not applicable.

**Informed Consent Statement:** Not applicable.

**Data Availability Statement:** Not applicable.

**Acknowledgments:** Not applicable.

**Conflicts of Interest:** The authors declare no conflict of interest.

## Appendix A. The Original Version of Questionnaire

Willingness to pay more

나는 친환경 드론 음식배달 서비스를 이용함에 있어서 추가적인 비용을 지불할 용의가 있다.

추가적인 비용을 발생하더라도 친환경 드론 음식배달 서비스를 이용할 의사가 있다.

나는 친환경 드론 음식배달 서비스를 이용하기 위해 추가적인 비용을 지불할 의사가 있다.

Word-of-mouth

나는 친환경 드론 음식배달 서비스를 다른 사람들에게 추천할 것 같다.

나는 다른 사람들에게 친환경 드론 음식배달 서비스를 이용하라고 권장할 것 같다.

나는 친환경 드론 음식배달 서비스와 관련해서 긍정적인 면을 다른 사람들에게 말할 것 같다.

Intentions to use

나는 집에서 배달 음식을 시킬 때 친환경 드론 음식배달 서비스를 이용할 의사가 있다.

나는 집에서 음식을 주문할 때 친환경 드론 음식배달 서비스를 이용할 것이다.

나는 집에서 음식을 배달시킬 때 친환경 드론 음식배달 서비스를 이용할 의지가 있다.

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
