# Peer review of "A Study on Behavioral Intentions in the Field of Eco-Friendly Drone Food Delivery Services: Focusing on Demographic Characteristics and Past Experiences"

_sustainability, doi:10.3390/su15076253_

Round 1
Reviewer 1 Report
The Hypothesis are not presenting something interesting and there are only 2 of them
The number of the participants is very low. Only a sample of 436 responses and only 237 of them have heard of drone delivery services.
The results present something that we are all expecting and there is not research interest.
Author Response
Reviewer 1
- The Hypothesis are not presenting something interesting and there are only 2 of them.
Response: Thank you for this important comment. As you suggested, we added hypotheses as follows:
☞ Hypothesis 1 (H1). There are differences with behavioral intentions based on the demographic characteristics.
Hypothesis 1a (H1a). There are differences with behavioral intentions based on the gender.
Hypothesis 1b (H1b). There are differences with behavioral intentions based on the age.
Hypothesis 1c (H1c). There are differences with behavioral intentions based on the level of education.
Hypothesis 1d (H1d). There are differences with behavioral intentions based on the marital status.
Hypothesis 1e (H1e). There are differences with behavioral intentions based on the level of monthly income.
Hypothesis 2 (H2). There are differences with behavioral intentions based on the past experiences.
Hypothesis 2a (H2a). There are differences with behavioral intentions based on whether respondents have heard of drone delivery services.
Hypothesis 2b (H2b). There are differences with behavioral intentions based on whether respondents have controlled a drone.
Hypothesis 2c (H2c). There are differences with behavioral intentions based whether respondents currently own a drone.
- The number of the participants is very low. Only a sample of 436 responses and only 237 of them have heard of drone delivery services.
Response: Thank you for pointing out this. As mentioned in the Limitation section, the drone delivery service has not been commercialized in Korea, so respondents do not have information about it. In addition, sample size is not one of the assumptions made in a t-test (Statology, 2022), so the samples used for the t-test do not seem to have any major problems.
Reference
Statology (2022). How to perform a t-test with unequal sample sizes. Retrieved from https://www.statology.org/t-test-unequal-sample-size/
- The results present something that we are all expecting and there is not research interest.
Response: Thank you for pointing out it. We extended discussion on the novelty of our study and the hypotheses rejected.
☞ First of all, the present paper is the first investigation of differences in behavioral intentions based on demographic factors and past experience. Although extant studies examined consumer behavior in the context of drone food delivery services [e.g. 4,13,29,39,41], they overlooked differences in behavioral intentions according to demographic characteristics, and there was also no study focusing on past experience. Thus, this study focused on demographic factors and past experience. The study developed the questionnaire by reviewing it with faculty members and drone experts, and successfully identified statistical differences in behavioral intentions based on demographic factors and past experiences.
More specifically, …
However, there are no statistical differences in behavioral intentions according to education level and whether consumers have owned a drone or not. Past studies stated that information and education about environmental protection can foster pro-environmental behavior [66]. As the issue of environmental pollution has become widely disseminated online, it might not difference in behavioral intentions according to education levels. In addition, consumers can directly/indirectly experience drones through YouTube videos, VR (virtual reality), and trial booths in the exhibition, even if they do not own a drone. Thus, there might be no difference in behavioral intentions based on whether or not they own a drone, unlike whether or not they had heard of or controlled a drone. These discussions imply that consumer behavior can change
Reviewer 2 Report
The authors tackle an interesting topic. The authors link eco-friendly with drones given that they use electric power rather than gasoline or other fossil fuel. This is an interesting approach. They carry out a survey with a decent data sample (422 individuals). I like that the 2 hypothesis are clearly stated and highlighted and the analysis seems to support the conclusions but I have a few comments:
In the abstract I think that the authors should include more of their results/major points. It is only mentioned that “there are statistical differences with the behavioral intentions in regards to the demographic characteristics” but they do not mention what are these differences.
The results could be explained a bit better. For example the authors mentioned that “females are more aware of environmental issues… more willing to pay extra for products”. I think that the authors have controlled for income level in the analysis but this is not clearly explained. Income can have a significant impact on this type of behavior.
There should be some comments about the skew in the demographics answering the survey. The proportion of educated females seems larger than in the overall population and there seems to be some skew in the age groups as well. This needs to be described with some more detail, particularly the potential impact of this in the results.
Related to the previous point. I think that it would be interesting to add a small description of the demographics in Korea and how they compare to the people answering the survey.
Table 4 has important content but the presentation makes it difficult to read. The authors should consider rearranging the table to make it more readable
In this type of survey the wording of the question is actually very important as it has been shown that people answer differently when asked basically the same question but with a different wording. In this regard, and for reproducibility purposes, it will be good if the authors could put in the supplementary documentation the actual questions (in Korean). No need to put it in the main body of the article but good to have them in the supplementary material
Author Response
Reviewer 2
The authors tackle an interesting topic. The authors link eco-friendly with drones given that they use electric power rather than gasoline or other fossil fuel. This is an interesting approach. They carry out a survey with a decent data sample (422 individuals). I like that the 2 hypothesis are clearly stated and highlighted and the analysis seems to support the conclusions but I have a few comments:
Response: Thank you for this encouragement and for all of your comments. Our responses to your comments are summarized in the following section.
- In the abstract I think that the authors should include more of their results/major points. It is only mentioned that “there are statistical differences with the behavioral intentions in regards to the demographic characteristics” but they do not mention what are these differences.
Response: Thank you for pointing out it. We added the result what are these differences.
☞ Abstract: Drones …. The data analysis results indicated that the females are more willing to pay extra than the males are, that the respondents who were in their 50s had higher word of mouth intention than the other generations, marital status showed significant differences in regards to willingness to pay more and intentions to use, and there was a difference with willingness to pay more and word-of-mouth in regards to monthly income. In addition, the respondents who had previously heard of drone food delivery services had higher averages with willingness to pay more and intentions to use as opposed to the respondents who had not heard of them, and the respondents who had experience controlling drones were willing to pay additional fees when they used eco-friendly drone food delivery services. …
- The results could be explained a bit better. For example the authors mentioned that “females are more aware of environmental issues… more willing to pay extra for products”. I think that the authors have controlled for income level in the analysis but this is not clearly explained. Income can have a significant impact on this type of behavior.
Response: Thank you for this significant comment. In this study, the level of monthly income is used as an independent variable rather than a control variable. In addition, as you mentioned, the level of monthly income has a significant impact on both willingness to pay more and word-of-mouth. Please see Table 4.
- There should be some comments about the skew in the demographics answering the survey. The proportion of educated females seems larger than in the overall population and there seems to be some skew in the age groups as well. This needs to be described with some more detail, particularly the potential impact of this in the results.
- Related to the previous point. I think that it would be interesting to add a small description of the demographics in Korea and how they compare to the people answering the survey.
Response: Thank you very much for valuable comments. Based on the Korean Statistical Information Service [62], gender and education level reflect South Korea's demographics, and the skew in age was written as a limitation of the study.
☞ 4.1. Respondent profile
Table 1 displays the demographic characteristics of the survey participants… According to Korean Statistical Information Service, there are more women than men, and the highest proportion of adults with a bachelor's degree in education [62], which is similar to the respondents' properties.
- Conclusions
… Nonetheless, the limitations of this study should not be overlooked. First, the generalizability of the findings is limited… In addition, the median age in South Korea is about 44 years old [62], so the results may not be representative of the country studied. It is necessary to design a sample that reflects the demographic population of the country studied.
Reference
- KOSIS. The population census in South Korea. Available online: https://kosis.kr/statHtml/statHtml.do?orgId=438&tblId=DT_438001_AA001&vw_cd=&list_id=00000184&scrId=&seqNo=&lang_mode=ko&obj_var_id=&itm_id=&conn_path=R1&path= (accessed on 28 March 2023)
- Table 4 has important content but the presentation makes it difficult to read. The authors should consider rearranging the table to make it more readable.
Response: Thank you for this important comment. We also tried to make Table 4 highly readable. As a result, Table 4 was created based on the paper previously published in the journal of Sustainability (e.g. Hwang et al., 2022; Kim, Joo, & Hwang, 2022). If the reviewer has a better way to think of it, please let us know. We will reflect your opinion 100%.
References
Hwang, J., Kim, H. M., Joo, K., Nawaz, M., & Moon, J. (2022). Travelers’ Perceived Value of Robot Services in the Airline Industry: Focusing on Demographic Characteristics. Sustainability, 14(23), 15818.
Kim, H. M., Joo, K., & Hwang, J. (2022). Are Customers Willing to Pay More for Eco-Friendly Edible Insect Restaurants? Focusing on the Internal Environmental Locus of Control. Sustainability, 14(16), 10075.
- In this type of survey the wording of the question is actually very important as it has been shown that people answer differently when asked basically the same question but with a different wording. In this regard, and for reproducibility purposes, it will be good if the authors could put in the supplementary documentation the actual questions (in Korean). No need to put it in the main body of the article but good to have them in the supplementary material.
Response: Thank you for this comment. We added the Korean questionnaire that was actually used during data collection to the Appendix.
☞ Appendix. The original version of questionnaire
Willingness to pay more
나는 친환경 드론 음식배달 서비스를 이용함에 있어서 추가적인 비용을 지불할 용의가 있다.
추가적인 비용을 발생하더라도 친환경 드론 음식배달 서비스를 이용할 의사가 있다.
나는 친환경 드론 음식배달 서비스를 이용하기 위해 추가적인 비용을 지불할 의사가 있다.
Word-of-mouth
나는 친환경 드론 음식배달 서비스와 관련해서 긍정적인 면을 다른 사람들에게 말할 것 같다.
나는 친환경 드론 음식배달 서비스를 다른 사람들에게 추천할 것 같다.
나는 다른 사람들에게 친환경 드론 음식배달 서비스를 이용하라고 권장할 것 같다.
Intentions to use
나는 집에서 배달 음식을 시킬 때 친환경 드론 음식배달 서비스를 이용할 의사가 있다.
나는 집에서 음식을 주문할 때 친환경 드론 음식배달 서비스를 이용할 것이다.
나는 집에서 음식을 배달 시킬 때 친환경 드론 음식배달 서비스를 이용할 의지가 있다.
Reviewer 3 Report
I appreciate the editor for providing me an opportunity to review the article “A Study on Behavioral Intentions in the Field of Eco-Friendly Drone Food Delivery Services: Focusing on Demographic Characteristics and Past Experiences”, which obviously falls in the scope of Sustainability. The paper is overall interesting and well-organized. In this informative article, the authors investigated the behavioral intentions of eco-friendly drone food delivery services.
1. I notice that the authors have published several research articles on the intentions of drone food delivery service, such as https://doi.org/10.1016/j.ijhm.2019.03.002. Considering the fact that the paper fails to explain the contribution, I would recommend the authors to conduct a comparison between this paper with the previous ones to better clarify its novelty.
2. In my opinion, the hypothesis should not be developed in section of literature review. A separate section of theoretical analysis is required.
3. I am confused of why the three factors derived by principal components analysis are named as willingness to pay more, word-of-mouth, and intentions to use. Please explain it.
4. I wonder if the t-tests are powerful enough to derive interesting conclusions. The results reported in Tables 4 and 5 can not adequately capture the interaction relationship between the three factors. What I expect is a regression model, instead of solely statistical test.
5. The directions of future research can be expanded.
Author Response
Reviewer3
I appreciate the editor for providing me an opportunity to review the article “A Study on Behavioral Intentions in the Field of Eco-Friendly Drone Food Delivery Services: Focusing on Demographic Characteristics and Past Experiences”, which obviously falls in the scope of Sustainability. The paper is overall interesting and well-organized. In this informative article, the authors investigated the behavioral intentions of eco-friendly drone food delivery services.
Response: Thank you for this encouragement and for all of your comments. Our responses to your comments are summarized in the following section.
- I notice that the authors have published several research articles on the intentions of drone food delivery service, such as https://doi.org/10.1016/j.ijhm.2019.03.002. Considering the fact that the paper fails to explain the contribution, I would recommend the authors to conduct a comparison between this paper with the previous ones to better clarify its novelty.
Response: Thank you for this critical comment. As you mentioned, there are some papers related to drone food delivery services. For example, the paper mentioned by the reviewer focused on the technical aspect of drone food delivery services, perceived innovativeness, but this study focused on the eco-friendly role of drone food delivery services. Following the reviewer’s comment, we tried to more clearly explain the novelty of our study.
☞ In addition, some papers have been done in the field of drone food delivery services. For example, Hwang, Kim, and Kim [23] applied the concept of motivated consumer innovativeness in drone food deliver services. Hwang, Kim, and Gulzar [24] examined how to form behavior intentions based on the theory of planned behavior in the context of drone food delivery services. More recently, Hwang, Kim, and Lee [25] investigated consumer innovativeness in the context of drone food delivery services. Unlike previous studies, this study explored the differences of behavioral intentions, such as intention to use, word-of-mouth, and wiliness to pay more with the demographic characteristics and the past experiences in the field of eco-friendly drone food delivery services for the first time. This is a significant difference from previous studies and can be said to be an important theoretical implication of this study.
References
- Hwang, J.; Kim, H.; Kim, W. Investigating motivated consumer innovativeness in the context of drone food delivery ser-vices. Journal of Hospitality and Tourism Management 2019, 38, 102-110.
- Hwang, J.; Kim, I.; Gulzar, M.A. Understanding the eco-friendly role of drone food delivery services: Deepening the theory of planned behavior. Sustainability 2020, 12, 1440.
- Hwang, J.; Kim, J.J.; Lee, K.W. Investigating consumer innovativeness in the context of drone food delivery services: Its impact on attitude and behavioral intentions. Technological Forecasting and Social Change 2021, 163, 120433.
- In my opinion, the hypothesis should not be developed in section of literature review. A separate section of theoretical analysis is required.
Response: According to your comment, we separated the section of literature review and hypothesis development.
☞ 3.2. Hypothesis development
Demographic characteristics have long been used in order to better understand consumer behavior in the field of social sciences [53-55]. Demographic characteristics in particular have an important meaning in regards to eco-friendly research. According to Han, Hsu, Lee, and Sheu [56], eco-friendly behavioral intentions are very different depending on the demographic characteristics of the consumers, so eco-friendly research must consider the demographic characteristics of the consumers. The existing empirical studies also show the importance of the demographic characteristics in eco-friendly research. For ex-ample, Kwon and Ahn [19] investigated the influence of the respondents’ demographic characteristics on behavioral intention toward green hotels. They discovered that age and education level are important factors that affect behavioral intention. Wang, Wong, and Narayanan [20] additionally explored the effect of the demographic characteristics of the consumers’ green purchase attitudes and behavioral intention in the green hotel industry. They suggested that there were significant differences with the green purchase attitudes in regards to age and income, whereas there was a significant difference with green behavioral intention in regards to education. Kim, Joo, and Hwang [57] more recently discovered that there were partial statistical differences of the four dimensions of the internal environmental locus of control, such as green consumers, environmental activists, environ-mental advocates, and recyclers in regards to gender, age, education, marital status, and monthly income.
Furthermore, past behavior shows the consumers’ habits and preferences, so identifying past behavior is very important in regards to understanding their behavior intentions [56]. It is particularly important to understand the past experiences in green research, because green behavior can be predicted from the past experiences [58]. The previous re-search also supports the significance of behavioral intentions. For example, Han and Hwang [59] developed a research model in order to find the role of the past experiences in the formation of behavioral intentions in an environmentally-responsible cruise context. They revealed that the past experiences affect behavioral intentions. Han, Hwang, Kim, and Jung [60] examined the effect of past behavior on pro-environmental intention to re-visit in a lodging context. They showed that past behavior is a key predictor of the pro-environmental intention to revisit. Han, Kim, and Lee [61] investigated how past experiences affect museum visitors’ pro-environmental behavior. They showed that the past experiences assist in order to form the museum visitors’ pro-environmental behavior. The following hypothesis is postulated as a result of the above arguments.
Hypothesis 1 (H1). There are differences with behavioral intentions based on the demographic characteristics.
Hypothesis 1a (H1a). There are differences with behavioral intentions based on the gender.
Hypothesis 1b (H1b). There are differences with behavioral intentions based on the age.
Hypothesis 1c (H1c). There are differences with behavioral intentions based on the level of education.
Hypothesis 1d (H1d). There are differences with behavioral intentions based on the marital status.
Hypothesis 1e (H1e). There are differences with behavioral intentions based on the level of monthly income.
Hypothesis 2 (H2). There are differences with behavioral intentions based on the past experiences.
Hypothesis 2a (H2a). There are differences with behavioral intentions based on whether respondents have heard of drone delivery services.
Hypothesis 2b (H2b). There are differences with behavioral intentions based on whether respondents have controlled a drone.
Hypothesis 2c (H2c). There are differences with behavioral intentions based whether respondents currently own a drone.
- I am confused of why the three factors derived by principal components analysis are named as willingness to pay more, word-of-mouth, and intentions to use. Please explain it.
Response: Thank you for this valuable comment. As mentioned in the manuscript, behavioral intentions included the three sub-dimensions, such as intentions to use, word-of-mouth, and willingness to pay more. In addition, the three sub-dimensions were not developed by us through measurement items, but rather by citing the dimensions developed by previous research. For this reason, we have named the three concepts willingness to pay more, word-of-mouth, and intentions to use. In addition, we explained how to measure the three dimensions in the ‘3.1. Measurement’ section as follows:
☞ Three items were borrowed from Zeithaml, Berry, and Parasuraman [50] in order to assess the intention to use. Word-of-mouth intentions were measured using three items that were obtained from Hennig-Thurau, Gwinner, and Gremler [51], and willingness to pay more was measured using three items from Han et al. [34].
References
- Han, H.; Hsu, L.T.J.; Lee, J.S. Empirical investigation of the roles of attitudes toward green behaviors, overall image, gender, and age in hotel customers’ eco-friendly decision-making process. International Journal of Hospitality Management 2009, 28, 519-528.
- Zeithaml, V.A.; Berry, L.L.; Parasuraman, A. The behavioral consequences of service quality. Journal of Marketing 1996, 60, 31-46.
- Hennig-Thurau, T.; Gwinner, K.P.; Gremler, D.D. Understanding relationship marketing outcomes: An integration of relational benefits and relationship quality. Journal of Service Research 2002, 4, 230-247.
- I wonder if the t-tests are powerful enough to derive interesting conclusions. The results reported in Tables 4 and 5 can not adequately capture the interaction relationship between the three factors. What I expect is a regression model, instead of solely statistical test.
Response: Thank you for the valuable comment. The purpose of this study was to examine the differences of behavioral intentions, including intention to use, word-of-mouth, and wiliness to pay more based on the demographic characteristics and the past experiences in the field of eco-friendly drone food delivery services. However, as your suggestion, it is important to find the antecedents of behavioral intentions using a regression model, so we have included this issue in the ‘Limitations’ section, and have mentioned the necessity of future studies.
☞ In addition, this study only focused on differences analysis as a one-way. ... It can also investigate the fac-tors predicting behavioral intentions, and perform regression analysis. For instance, the risks that the consumers are concerned about with the services weakened their behavioral intentions [13,29]. On the other hand, the individuals’ innovativeness strengthens their behavioral intentions [15]. The present study might have overlooked the effects of these types of variables, so it suggests further research design a framework that encompasses both negative and positive factors with demographic factors.
- The directions of future research can be expanded.
Response: Thanks for your helpful comment. We revised and expanded future research about limitations.
☞ … Nonetheless, the limitations of this study should not be overlooked. First, the generalizability of the findings is limited, because the present study collected data from only Korean respondents. In addition, the median age in South Korea is about 44 years old [62], so the results may not be representative of the country studied. It is necessary to design a sample that reflects the demographic population of the country studied. Also, it was not the actual users of drone food delivery services, because they have not yet been fully commercialized in Korea. The future studies should consider the data-collecting method, which can improve the generalizability, such as cross-cultural studies and research on the actual users. In addition, this study only focused on differences analysis as a one-way. Algorithms of answertree can provide in-depth results by splitting nodes. It can also investigate the factors predicting behavioral intentions, and perform regression analysis. For instance, the risks that the consumers are concerned about with the services weakened their behavioral intentions [13,29]. On the other hand, the individuals’ innovativeness strengthens their behavioral intentions [15]. The present study might have overlooked the effects of these types of variables, so it suggests further research design a framework that encompasses both negative and positive factors with demographic factors. Lastly, the study only focused on the consumers’ personal conditions, which included the demographic factors and the past experiences. Behavioral intentions can be fostered by social factors, such as norms and social influence [4,36]. The study suggests that the future research consider the differences in social motives in the context of drone food delivery services.
Reviewer 4 Report
The paper is an interesting study on how potential consumer assess eco-friendly food delivery services realized by drones. In my opinion this is an innovative and attractive topic.
The authors aimed at examining the differences with the behavioral intentions, which include intention to use, word-of-mouth, and willingness to pay more with the demographic characteristics and the past experiences in the context of eco-friendly drone food delivery services.
The design of the study, theoretical background, clarity of thought, concluding are on a good level in my opinion. Also statistical procedures, encompassing exploratory factor analysis (PCA), ANOVA and other test are well chosen and used.
I would however suggest a few improvements for this paper:
1) The paper lacks a traditional point "Discussion". Some of discussion with other authors is present in "Implications" and "Conclusions", but this is rather limited. Please explore discussion in some more intensive matter.
2) The paper also lacks points: "Research limitations" and "Further research". They should not be long, however I think they are needed, as they point out that the Authors are aware of the limitations of their research and also they find it purposive to continue the study in a selected direction.
Author Response
Reviewer 4
The paper is an interesting study on how potential consumer assess eco-friendly food delivery services realized by drones. In my opinion this is an innovative and attractive topic. The authors aimed at examining the differences with the behavioral intentions, which include intention to use, word-of-mouth, and willingness to pay more with the demographic characteristics and the past experiences in the context of eco-friendly drone food delivery services. The design of the study, theoretical background, clarity of thought, concluding are on a good level in my opinion. Also statistical procedures, encompassing exploratory factor analysis (PCA), ANOVA and other test are well chosen and used. I would however suggest a few improvements for this paper:
Response: Thank you for this encouragement and for all of your comments. Our responses to your comments are summarized in the following section.
- The paper lacks a traditional point "Discussion". Some of discussion with other authors is present in "Implications" and "Conclusions", but this is rather limited. Please explore discussion in some more intensive matter.
Response: Thank you for pointing out it. We extended discussion on the novelty of our study and the hypotheses rejected.
☞ First of all, the present paper is the first investigation of differences in behavioral intentions based on demographic factors and past experience. Although extant studies examined consumer behavior in the context of drone food delivery services [e.g. 4,13,29,39,41], they overlooked differences in behavioral intentions according to demographic characteristics, and there was also no study focusing on past experience. Thus, this study focused on demographic factors and past experience. The study developed the questionnaire by reviewing it with faculty members and drone experts, and successfully identified statistical differences in behavioral intentions based on demographic factors and past experiences.
More specifically, …
However, there are no statistical differences in behavioral intentions according to education level and whether consumers have owned a drone or not. Past studies stated that information and education about environmental protection can foster pro-environmental behavior [66]. As the issue of environmental pollution has become widely disseminated online, it might not difference in behavioral intentions according to education levels. In addition, consumers can directly/indirectly experience drones through YouTube videos, VR (virtual reality), and trial booths in the exhibition, even if they do not own a drone. Thus, there might be no difference in behavioral intentions based on whether or not they own a drone, unlike whether or not they had heard of or controlled a drone. These discussions imply that consumer behavior can change
- The paper also lacks points: "Research limitations" and "Further research". They should not be long, however I think they are needed, as they point out that the Authors are aware of the limitations of their research and also they find it purposive to continue the study in a selected direction.
Response: Thanks for your helpful comment. We revised and expanded future research about limitations.
☞ … Nonetheless, the limitations of this study should not be overlooked. First, the generalizability of the findings is limited, because the present study collected data from only Korean respondents. In addition, the median age in South Korea is about 44 years old [62], so the results may not be representative of the country studied. It is necessary to design a sample that reflects the demographic population of the country studied. Also, it was not the actual users of drone food delivery services, because they have not yet been fully commercialized in Korea. The future studies should consider the data-collecting method, which can improve the generalizability, such as cross-cultural studies and research on the actual users. In addition, this study only focused on differences analysis as a one-way. Algorithms of answertree can provide in-depth results by splitting nodes. It can also investigate the factors predicting behavioral intentions, and perform regression analysis. For instance, the risks that the consumers are concerned about with the services weakened their behavioral intentions [13,29]. On the other hand, the individuals’ innovativeness strengthens their behavioral intentions [15]. The present study might have overlooked the effects of these types of variables, so it suggests further research design a framework that encompasses both negative and positive factors with demographic factors. Lastly, the study only focused on the consumers’ personal conditions, which included the demographic factors and the past experiences. Behavioral intentions can be fostered by social factors, such as norms and social influence [4,36]. The study suggests that the future research consider the differences in social motives in the context of drone food delivery services.
Round 2
Reviewer 1 Report
My review requests has been improved.
Ηowever, I suggest in a future research to apply simulation and modelling, which can show the evolution of a process in the future.
In this case, it could be seen in the future the mood of the users to increase or decrease the use of drones